# Surface Vision Transformers: Attention-Based Modelling applied to Cortical Analysis

**Simon Dahan**[1]                                    SIMON.DAHAN@KCL.AC.UK

**Abdulah Fawaz**[1]                                  ABDULAH.FAWAZ@KCL.AC.UK

**Logan Z. J. Williams**[1,2]                         LOGAN.WILLIAMS@KCL.AC.UK

**Chunhui Yang**[3]                                   CHUNHUIYANG@WUSTL.EDU

**Timothy S. Coalson**[3]                             COALSONT@WUSTL.EDU

**Matthew F. Glasser**[4]                             GLASSERM@WUSTL.EDU

**A. David Edwards**[2]                               AD.EDWARDS@KCL.AC.UK

**Daniel Rueckert**[5]                                D.RUECKERT@IMPERIAL.AC.UK

**Emma C. Robinson**[1,2]                             EMMA.ROBINSON@KCL.AC.UK

[1] *Department of Biomedical Engineering, School of Biomedical Engineering and Imaging Sciences, King's College London.*

[2] *Centre for the Developing Brain, Department of Perinatal Imaging and Health, School of Biomedical Engineering and Imaging Sciences, King's College London.*

[3] *Department of Radiology, Washington University Medical School.*

[4] *Department of Neuroscience, Washington University Medical School.*

[5] *Biomedical Image Analysis Group, Department of Computing, Imperial College London.*

**Editors:** Under Review for MIDL 2022

## Abstract

The extension of convolutional neural networks (CNNs) to non-Euclidean geometries has led to multiple frameworks for studying manifolds. Many of those methods have shown design limitations resulting in poor modelling of long-range associations, as the generalisation of convolutions to irregular surfaces is non-trivial. Motivated by the success of attention-modelling in computer vision, we translate convolution-free vision transformer approaches to surface data, to introduce a domain-agnostic architecture to study any surface data projected onto a spherical manifold. Here, surface patching is achieved by representing spherical data as a sequence of triangular patches, extracted from a subdivided icosphere. A transformer model encodes the sequence of patches via successive multi-head self-attention layers while preserving the sequence resolution. We validate the performance of the proposed Surface Vision Transformer ($SiT$) on the task of phenotype regression from cortical surface metrics derived from the Developing Human Connectome Project (dHCP). Experiments show that the $SiT$ generally outperforms surface CNNs, while performing comparably on registered and unregistered data. Analysis of transformer attention maps offers strong potential to characterise subtle cognitive developmental patterns.

**Keywords:** Vision Transformer, Cortical Analysis, Deep Learning, Neuroimaging, Attention-based Modelling.

## 1. Introduction

Over recent years, attention-based transformer models have dominated the field of natural language processing (NLP) through supporting the learning of long-distance associations within sequences (Vaswani et al., 2017; Radford et al., 2018; Devlin et al., 2019). Since thorough understanding of context within images is essential for most computer vision task, Dosovitskiy et al. 2020 proposed the vision transformer ($ViT$) to adapt NLP transformer architectures (Vaswani et al., 2017) to the natural imaging domain, by processing images as sequences of patches. In doing so, they showed that pure transformer models were capable of outperforming CNNs for image classification, while addressing some of their limitations in terms of their inductive bias towards local features, lack of scalability and low parameter-efficiency.

As there is no generic deep learning method for studying non-Euclidean data, there has also been an emerging interest in translating attention-based mechanisms to irregular geometries, as a way to improve the learning of long-range associations and feature expressiveness. The transformer architecture has been used with some successes in graph domain (Yang et al., 2021; Dwivedi and Bresson, 2021), or for sequences of 3D meshes and point-clouds (Sarasua et al., 2021; Zhao et al., 2021; Guo et al., 2021), while He et al. 2021 introduced a self-attention mechanism for general manifolds to ensure gauge equivariance. However, to date, there has been no application of vision transformers to generic functions on surfaces, which are an important data structure across many distinct disciplines (Boomsma and Frellsen, 2017; Jiang et al., 2019; Defferrard et al., 2020), particularly in medical imaging (Gopinath et al., 2019; Kong and Shadden, 2021; Ma et al., 2021; Lebrat et al., 2021).

In this paper, we therefore seek to adapt the data efficient image transformer ($DeiT$) model (Touvron et al., 2020) to surface domains by projecting data onto a sphere and patching surfaces using a regular icospheric tessellation. The methodology of surface patching is general and the proposed $SiT$ may be adapted for any genus-zero surface. One challenging area, which stands to particularly benefit from this style of analysis is the study of the cerebral cortex, since traditional approaches for brain image analysis based on registration, have historically been unable to fully capture the heterogeneity of cortical organisation across individuals, not only for vulnerable groups (Ciarrusta et al., 2020), but also within healthy populations (Fischl et al., 2008; Frost and Goebel, 2012; Glasser et al., 2016; Kong et al., 2019). We therefore evaluate our approach by comparing our model against a range of convolutional geometric deep learning frameworks on the task of developmental phenotype regression from cortical metric data derived from the Developing Human Connectome Project (dHCP).[1] The main contributions of this work can be summarised as follows:

- This paper proposes translation of vision transformers (Dosovitskiy et al., 2020; Touvron et al., 2020) to any data associated with genus-zero surfaces.

- Validation of the model on the task of neurodevelopmental phenotype regression demonstrates competitive performance relative to spectral and spatial geometric deep learning frameworks (Cohen et al., 2018; Monti et al., 2016; Defferrard et al., 2017; Kipf and Welling, 2017; Zhao et al., 2019).

---

1. The code is available at https://github.com/metrics-lab/surface-vision-transformers

- Visualisation of attention weights highlights that the model attends to well-characterised spatiotemporal patterns of perinatal cortical development (Dubois et al., 2008; Doria et al., 2010; Eyre et al., 2021).

## 2. Related work

**Attention-based modelling and Transformers.** Attention was first introduced by Bahdanau et al. 2016 as a tool for modelling long-range dependencies, in recurrent networks trained for machine translation tasks. To increase context modelling in long-sequences, Vaswani et al. 2017 introduced the self-attention mechanism - learning attention between elements in a sequence - with the transformer architecture. When paired with pre-training on very large datasets, this led to powerful language models such as BERT or GPT (Devlin et al., 2019; Radford et al., 2018), that are transferable to many downstream tasks. While CNNs have proven to be sample-efficient architectures for image understanding, most architectures saturate in performance when presented with very large datasets, and struggle to relate distant elements within images. This is partly due to the inherent inductive biases (locality and translational equivariance) of the CNN architecture. As a result, inspired by the success of attention mechanisms, efforts have been made to incorporate attention-based operations into CNNs (Jaderberg et al., 2016; Wang et al., 2018a; Hu et al., 2019), including for medical imaging (Wang et al., 2018b; Schlemper et al., 2019), as a way to increase the learning of contextual information. However, these have been limited by the heavy cost of pixel-level attention and so computation was mostly restricted to low-resolution feature maps. More recently, vision transformer architectures have been introduced to adapt the sequence-modelling of NLP in the context of computer vision. Dosovitskiy et al. 2020 introduced a patching strategy consisting of splitting RGB images into patches of shape $16 \times 16 \times 3$. By doing so, the attention is computed at the patch level, and contextual information from the entire image is available already from early layers. These have been shown to outperform CNNs for a range of computer vision tasks including image classification and segmentation (Dosovitskiy et al., 2020; Touvron et al., 2020; d'Ascoli et al., 2021), object detection (Xu et al., 2021) and video classification (Liu et al., 2021), achieving state-of-the-art performance without relying on strong spatial priors. In medical imaging tasks, modelling long-range dependencies is critical, for instance for anomaly detection or image segmentation, but limited by most of the current convolution-based models. Therefore, hybrid and pure vision transformers have also been translated with success for medical image segmentation (Pinaya et al., 2021; Chen et al., 2021a; Karimi et al., 2021; Zhang et al., 2021a; Gao et al., 2021), medical image registration (Chen et al., 2021b; Zhang et al., 2021b; van Tulder et al., 2021) or medical image reconstruction (Feng et al., 2021).

**Geometric Deep Learning.** A prominent approach for studying cortical surfaces relies on geometric deep learning (gDL) methods (Gopinath et al., 2019; Arya et al., 2020; Kim et al., 2021) that aim to adapt Euclidean CNNs to irregular manifolds (Bronstein et al., 2016, 2021). While many frameworks for surface and graph convolutions exist, typically these frameworks struggle to learn rotationally equivariant, expressive features, without prohibitively high computational cost (Bruna et al., 2013; Cohen et al., 2018). These limitations have been stressed in a recent benchmarking paper (Fawaz et al., 2021), which evaluated geometric deep learning techniques in the context of cortical analysis.

## 3. Methods

### 3.1. Data

Data in this work comes from the publicly available third release of the Developing Human Connectome Project (dHCP)[2] (Hughes et al., 2017) and contains cortical surface meshes and metrics (sulcal depth, curvature, cortical thickness and T1w/T2w myelination) derived from T1 and T2-weighted magnetic resonance images (MRI), processed according to Makropoulos et al. 2018 and references therein (Kuklisova-Murgasova et al., 2012; Schuh et al., 2017; Hughes et al., 2017; Cordero-Grande et al., 2018; Makropoulos et al., 2018). We included a total of 588 images acquired from term (born $\geq 37$ weeks gestational age, GA) and preterm ($< 37$ weeks GA) neonatal subjects, scanned between 24 and 45 weeks postmenstrual age (PMA). Some of the preterm neonates were scanned twice: once after birth and again around term-equivalent age.

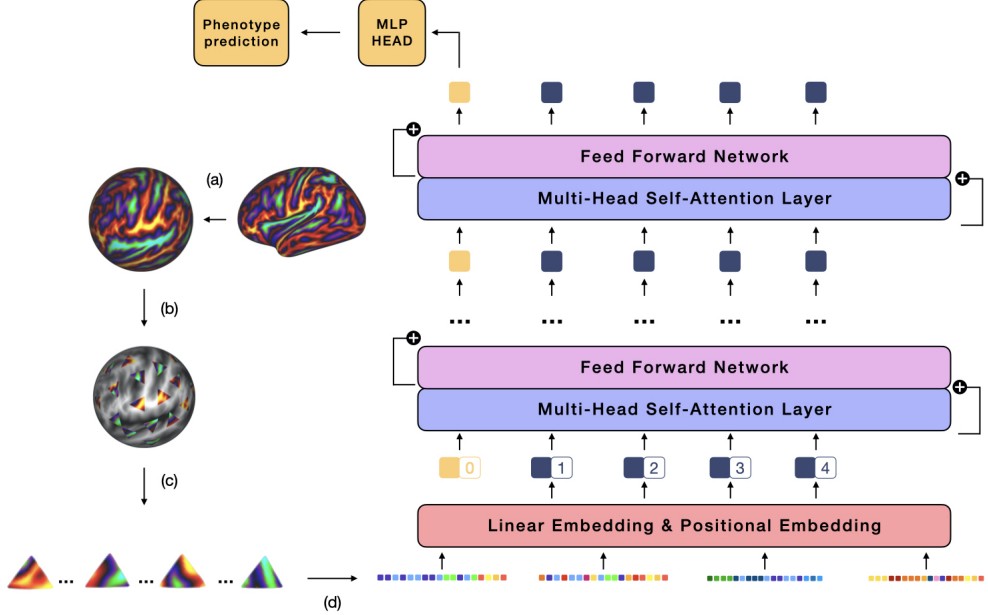

Figure 1: Surface Vision Transformer ($SiT$) architecture. The cortical data is first resampled, using barycentric interpolation, from its template resolution (32492 vertices) to a sixth order icosphere (mesh of 40962 equally spaced vertices). The regular icosphere is divided into triangular patches of equal vertex count (b, c) that fully cover the sphere (not shown), which are flattened into feature vectors (d), and then fed into the transformer model.

The proposed framework was benchmarked on two phenotype regression tasks: prediction of postmenstrual age (PMA) at scan, and gestational age (GA) at birth. Since the objective is to model PMA and GA as markers of healthy development, all preterms' second scans were excluded from the PMA prediction task, and their first scans were excluded from the GA regression task. This resulted in 530 neonatal subjects for the PMA prediction task (419 term/111 preterm), and 514 neonatal subjects (419 term/95 preterm) for the GA

---

2. http://www.developingconnectome.org.

| Models | Layers | Heads | Hidden size $D$ | MLP size | Params. |
|--------|--------|-------|-----------------|----------|---------|
| SiT-tiny | 12 | 3 | 192 | 768 | 5M |
| SiT-small | 12 | 6 | 384 | 1536 | 22M |
| SiT-base | 12 | 12 | 768 | 3072 | 86M |

Table 1: All *SiT* models preserve a hidden size of 64 per attention head.

prediction task. In all instances, the proposed transformer networks are compared against the best performing surface CNNs reported in Fawaz et al. 2021: Spherical UNet (Zhao et al., 2019), MoNet (Monti et al., 2016), GConvNet (Kipf and Welling, 2017), ChebNet (Defferrard et al., 2017) and S2CNN (Cohen et al., 2018); on cortical surface data following the same pre-processing pipeline as in Fawaz et al. 2021. Therefore, experiments were run on both *template*-aligned data and non-registered (*native*) data, and train/test/validation splits parallel those used in Fawaz et al. 2021. See Appendix A.3 for more details on image preprocessing and registration pipelines, and Appendix A.4 for training details of gDL methods.

### 3.2. Surface Vision Transformer

**Architecture.** The proposed *SiT* model builds upon three variants of the data efficient image transformer or *DeiT* (Touvron et al., 2020): *DeiT-Tiny*, *DeiT-Small*, *DeiT-Base*. For more details about the *SiT* model architectures see Table 1. In general terms, for any vision transformer, the high-resolution grid of the input domain $X$, is reshaped into a sequence of $N$ flattened patches $\widetilde{X} = \left[\widetilde{X}_1^{(0)}, ..., \widetilde{X}_N^{(0)}\right] \in \mathbb{R}^{N \times (VC)}$ (V vertices, C channels). This initial sequence is first projected onto a sequence of dimension $D$ with a trainable linear layer; an extra token for regression is concatenated to the patch sequence, then a positional embedding is added to the patch embeddings, such that the input sequence of the transformer is $X^{(0)} = \left[X_0^{(0)}, ..., X_N^{(0)}\right] \in \mathbb{R}^{(N+1) \times D}$ (see details in Appendix C.2). For the *SiT*, this is implemented by imposing a low-resolution triangulated grid, on the input mesh, using a regularly tessellated icosphere (Fig 1). This generates 320 patches, per hemisphere, and per channel. In all cases, before extracting patches, the right hemisphere is flipped to mirror the left orientation. Since there are four input channels (myelin, sulcal depth, curvature and cortical thickness) and the number vertices per patch is 153, the total dimension per patch is $VC = 612$. The architecture is made of $L$ transformer blocks, each composed of a multi-head self-attention layer (MSA), implementing the self-attention mechanism across the sequence, and a feed-forward network (FFN), which expands then reduces the sequence dimension (see details in Appendix C.1). In short, for every transformer block at layer $l$, the input sequence $X^{(l)} = \left[X_0^{(l)}, ..., X_N^{(l)}\right]$ is processed as follows:

$$\begin{aligned}
Z^{(l)} &= \boldsymbol{MSA}(X^{(l)}) + X^{(l)} \\
X^{(l+1)} &= \boldsymbol{FFN}(Z^{(l)}) + Z^{(l)} \\
&= \left[X_0^{(l+1)}, ..., X_N^{(l+1)}\right] \in \mathbb{R}^{(N+1) \times D}
\end{aligned} \tag{1}$$

Following standard practice in transformers, *LayerNorm* (Ba et al., 2016) is used prior to each MSA and FFN layer (omitted for clarity in Eq 1) and residual connections are

added thereafter. The MSA used in transformers runs multiple self-attention in parallel at each layer, referred as *Heads* in Table 1. In practice, it means that the input sequence of dimension $D$ is divided into sub-parts of dimension $D_h = D/h$. After being run in parallel, the self-attention heads are concatenated. The regression patch of the final sequence $X^{(L)}$ is used as input of an MLP head for prediction (see Fig 1).

**Attention maps.** Unlike CNNs, the vision transformer model learns associations between patches within a sequence thanks to a self-attention mechanism inherited from NLP Transformer (Vaswani et al., 2017). Attention maps can be visualised on the input space by propagating the attention weights across layers and heads from the regression token of the last transformer block (details in Appendix C.3). As each transformer head is running in parallel, they should attend to different parts of the input sequence.

**Pre-training.** Previous works pointed the necessity to pre-train vision transformers on large-scale datasets to mitigate the lack of inductive biases in the architecture (Dosovitskiy et al., 2020; Steiner et al., 2021; Beal et al., 2021). Lately, the need for large pretraining has been reviewed as alternative training strategies such as Knowledge Distillation have emerged (Touvron et al., 2020). This is relevant for medical imaging tasks, as datasets are usually smaller than in natural imaging, and can benefit from pretraining before transferring to downstream tasks. Therefore, in this paper we evaluate different training strategies, to explore: 1) training from scratch; 2) initialising from ImageNet weights (to support training on small datasets through incorporation of some spatial priors) and 3) fine-tuning after a self-supervision learning pretraining task (SSL). For ImageNet, we used pretrained models from the **timm** open-source library[3], where models were pretrained on ImageNet2012 (1 million images, 1000 classes) on patches of size $16 \times 16 \times 3$. For self-supervision, we adapted the *masked patch prediction* task (MPP) for self-supervision used in BERT (Devlin et al., 2019), which consists of corrupting at random some input patches in the sequence; then training the network to learn how to reconstruct the full corrupted patches. In this setting, we corrupt at random 50% of the input patches, either replacing them with a learnable mask token (80%), another patch embedding from the sequence at random (10%) or keeping their original embeddings (10%). To optimise the reconstruction, the mean square error (MSE) loss is computed only for the patches in the sequence that were masked. Pretraining was performed with template (registered) training data only, as adding unregistered data to the pre-training did not seem to improve results (see Appendix B.1).

## 4. Results & Discussion

Test results for the tasks of postmenstrual age (PMA) at scan and gestational age (GA) at birth are reported in Table 2 for the three training strategies: training from scratch, from ImageNet weights and from SSL weights and compared against best gDL methods in Fawaz et al. 2021. Overall $SiT$ models consistently outperformed two of the gDL methods (ChebNet and GConvNet) and were competitive compared to the three best performing geometric models (S2CNN, Spherical UNet and MoNet). Spherical UNet obtained excellent performances on template space for both tasks in Fawaz et al. 2021, but greatly underperformed on native space, since it builds no transformation equivariance into its model. All $SiT$

---

3. pretrained models on ImageNet available at http://github.com/rwightman/pytorch-image-models/

| Methods | ImageNet | MPP | PMA | | | GA - deconfounded | | | Avg |
|---|---|---|---|---|---|---|---|---|---|
| | | | Template | Native | Avg | Template | Native | Avg | |
| S2CNN | ✗ | ✗ | 0.63±0.02 | 0.73±0.25 | 0.68 | 1.35±0.68 | 1.52±0.60 | 1.44 | 1.06 |
| ChebNet | ✗ | ✗ | 0.59±0.37 | 0.77±0.49 | 0.68 | 1.57±0.15 | 1.70±0.36 | 1.64 | 1.16 |
| GConvNet | ✗ | ✗ | 0.75±0.13 | 0.75±0.26 | 0.75 | 1.77±0.26 | 2.30±0.74 | 2.04 | 1.39 |
| Spherical UNet | ✗ | ✗ | 0.57±0.18 | 0.87±0.50 | 0.72 | **0.85±0.17** | 2.16±0.57 | 1.51 | 1.11 |
| MoNet | ✗ | ✗ | 0.57±0.02 | **0.61±0.05** | **0.59** | 1.44±0.08 | 1.58±0.06 | 1.51 | 1.05 |
| SiT-tiny | ✗ | ✗ | 0.63±0.01 | 0.77±0.03 | 0.70 | 1.37±0.03 | 1.66±0.06 | 1.52 | 1.11 |
| SiT-tiny | ✓ | ✗ | 0.67±0.02 | 0.70±0.04 | 0.69 | 1.22±0.05 | 1.69±0.05 | 1.46 | 1.07 |
| SiT-tiny | ✗ | ✓ | 0.58±0.01 | 0.64±0.06 | 0.61 | 1.18±0.07 | 1.61±0.03 | 1.39 | 1.00 |
| SiT-small | ✗ | ✗ | 0.60±0.02 | 0.76±0.03 | 0.68 | 1.14±0.12 | **1.44±0.03** | **1.29** | 0.99 |
| SiT-small | ✓ | ✗ | 0.59±0.03 | 0.71±0.02 | 0.65 | 1.15±0.05 | 1.69±0.03 | 1.42 | 1.04 |
| SiT-small | ✗ | ✓ | **0.55±0.04** | 0.63±0.06 | **0.59** | 1.13±0.02 | 1.47±0.08 | 1.30 | **0.95** |
| SiT-base | ✗ | ✗ | 0.59±0.01 | 0.68±0.03 | 0.64 | 1.12±0.10 | 1.46±0.11 | **1.29** | 0.96 |
| SiT-base | ✓ | ✗ | 0.61±0.04 | 0.75±0.01 | 0.68 | 1.05±0.11 | 1.52±0.08 | **1.29** | 0.98 |
| SiT-base | ✗ | ✓ | 0.61±0.04 | 0.70±0.03 | 0.66 | 0.97±0.07 | 1.61±0.08 | **1.29** | 0.97 |

Table 2: Best **M**ean **A**bsolute **E**rror (in weeks) and standard deviations for the three best models are reported; *SiT-tiny*, *SiT-small*, and *SiT-base* are compared against surface CNN frameworks: S2CNN, ChebNet, GConvNet, Spherical Unet and MoNet. Average results per task and overall are displayed. Training details and experimental set-up in A.1, A.2.

models achieved similar performance to Spherical UNet on template, with improvements on PMA (0.55 for *SiT-small*), but dropped less in performance between template/native predictions: for instance *SiT-tiny* achieved 0.58/0.64 on PMA against 0.57/0.87 for Spherical UNet. This seems to indicate some robustness to transformation. Generally, *SiT-tiny* and *SiT-small* performed fairly well and consistently on both tasks, especially on finetuned models following self-supervision. Results would suggest the need of regularisation for *SiT-base* (especially dropout) whereas *SiT-small* achieved the best performance on average (template & native) for both tasks. Overall, almost all configurations of *SiT* achieved better average performances (on both tasks) than gDL methods, that were optimised with various data augmentation techniques in Fawaz et al. 2021.

**Birth age task.** The task of GA prediction is arguably more complicated than the PMA task, as it is run on scans acquired around term-equivalent age ($\geq 37$ weeks GA) for both term and preterm neonates, and therefore is highly correlated to PMA at scan. Therefore, a deconfounding strategy of PMA at scan for the task of birth age prediction was employed and described in details in Appendix B.2. *SiT* models achieved good performances without major drops in performance between template and native for this task, a trend that can be seen across *SiT* versions. For details of deconfounding of gDL methods see Appendix A.4.

**Training methods.** For the task of PMA at scan, while comparing the three training strategies, training from ImageNet often underperformed training from scratch, while training after the MPP task achieved the best performance with *SiT-tiny* and *SiT-small*. Self-supervision pretraining appeared to improve training quality, for nearly all configurations and both tasks. This is consistent with results in natural imaging. As a vision transformer is a more general (less inductive biases) architecture than a CNN, it seems natural that pretraining is beneficial to compensate for the lack of geometric prior.

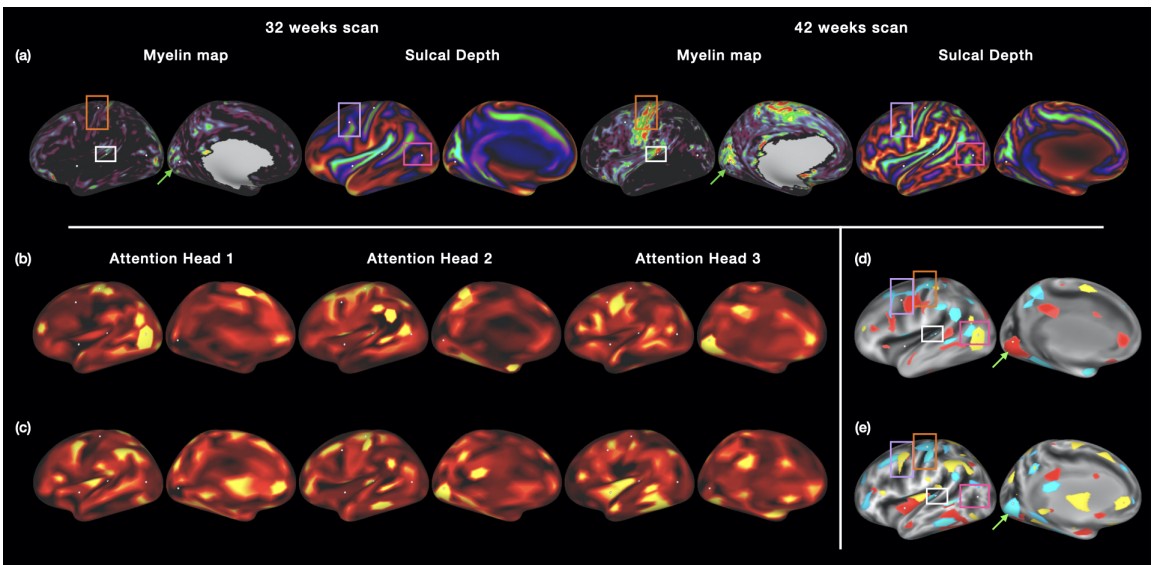

Figure 2: Example of attention maps for a single preterm neonate (born at 26.7 weeks GA) scanned at multiple time points (32.4 and 41.9 weeks PMA). T1w/T2w ratio and sulcal depth maps are shown in (a), and the attention maps for the 32 week and 42 week scans are shown in (b) and (c), respectively. (d) and (e) represent the thresholded attention maps for the 32 week and 42 week scans, respectively. The attention maps capture differences in the T1w/T2w ratio of the somatosensory (orange box), auditory (white box) and visual (green arrow) cortices, and changes in sulcal depth in the frontal (purple box) and temporo-occipital (magenta box) cortices. Additional illustrations are provided in Appendix C.3.

**Attention Maps.** Figure 2 shows the attention per head for a preterm-born neonate (born at 26.7 weeks) that was scanned twice: once at 32.4 weeks and a second time at 41.9 weeks. Attention maps captured the well-characterised spatio-temporal patterns of perinatal brain development. Maturation of cortical regions follows a primary-to-association trajectory (Doria et al., 2010; Eyre et al., 2021), highlighted by increases in T1w/T2w myelin contrast in the visual, auditory and somatosensory areas, as well as rapid cortical folding over the third trimester, particularly in the parietal and frontal lobes (Dubois et al., 2008).

## 5. Conclusion

In this paper we translated the methodology of vision transformers to surfaces and validated on the challenging task of phenotype prediction from cortical imaging data. Vision transformers offer strong potential for analysing signals, such as cortical or cardiac, on surface, as they show robustness to transformations and can integrate information between distant regions. Future work could focus on improving training strategies, as the largest $SiT$ architectures suffer from the limited size of medical datasets and could benefit from data augmentation, regularisation and alternative self-supervision pretraining strategies (Touvron et al., 2019; Kolesnikov et al., 2019).

## Acknowledgments

We would like to acknowledge funding from the EPSRC Centre for Doctoral Training in Smart Medical Imaging (EP/S022104/1). Data were provided by the developing Human Connectome Project, KCL-Imperial-Oxford Consortium funded by the European Research Council under the European Union Seventh Framework Programme (FP/2007-2013) / ERC Grant Agreement no. [319456]. We are grateful to the families who generously supported this trial. CY/TSC/MFG were supported by NIH R01 MH060974.

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

## Appendix A. Training details

### A.1. Experimental set-up & training strategy

All experiments were run on a single NVIDIA TITAN RTX or RTX 3090 24GB GPU. The batch size was maximised to use all GPU memory at training time (256 for *SiT-Tiny*, 128 for *SiT-Small* and 64 for *SiT-Base*). The training strategy used for each task and model is summarised in Table 3. For trainings from ImageNet or MPP weights, models were finetuned for 1000 epochs, as their convergence was faster. Models were optimised similarly for native and template space. Linear warm-up was used for 50 epochs. MSE loss is used for optimisation.

| Model | Task | Optimiser | Warm-up | | Learning Rate | | Training Iterations | |
|-------|------|-----------|---------|------------|---------------|------------|---------------------|------------|
| | | | Scratch | Finetuning | Scratch | Finetuning | Scratch | Finetuning |
| SiT-tiny | PMA | SGD | ✓ | ✗ | $1e^{-4}$ | $1e^{-5}$ | 2000 | 1000 |
| SiT-small | PMA | SGD | ✓ | ✗ | $1e^{-4}$ | $1e^{-5}$ | 2000 | 1000 |
| SiT-base | PMA | SGD | ✓ | ✗ | $1e^{-4}$ | $1e^{-5}$ | 2000 | 1000 |
| SiT-tiny | GA | Adam | ✓ | ✗ | $5e^{-4}$ | $3e^{-4}$ | 2000 | 1000 |
| SiT-small | GA | Adam | ✓ | ✗ | $5e^{-4}$ | $3e^{-4}$ | 2000 | 1000 |
| SiT-base | GA | Adam | ✗ | ✗ | $1e^{-4}$ | $5e^{-4}$ | 2000 | 1000 |

Table 3: Training strategies for all models and task. Finetuning refers to models trained from ImageNet weights or after self-supervision (MPP).

### A.2. Hyperparameter search & training strategy

A hyperparameter search was run for the PMA and GA (Table 4) prediction tasks. Hyperparameters were optimised for the *SiT-tiny* model and include choice of optimiser and best learning strategy, where this includes investigating use of a scheduler, and performing warm up. For the task of PMA prediction, the hyperparameter search suggested best use of SGD with warm-up; whereas for GA, best models were obtained using Adam.

### A.3. Pre-processing

Surface data used were generated by the dHCP structural pipeline (Makropoulos et al., 2018). Briefly, motion-corrected, reconstructed T2w and T1w images were bias corrected and brain extracted. Next, images were segmented into several tissue types using the Draw-EM algorithm. Following this, topologically correct white matter surfaces were fit to the grey-white tissue boundary. Pial surfaces were then generated by expanding the white matter mesh towards the grey-cerebrospinal fluid boundary. Inflated surfaces (including spheres) were generated through an iterative process of inflation and smoothing. The dHCP structural pipeline generated a number of univariate features surface-based features summarising cortical organisation. In this paper, we used: sulcal depth, curvature, cortical thickness (estimated as the Euclidean distance between corresponding vertices in the white and pial surfaces), and the T1w/T2w ratio maps, which are highly correlated with intracortical myelination (Glasser and Van Essen, 2011; Soun et al., 2017).

Spherical-based surface registration was performed using Multimodal Surface Matching (Robinson et al., 2014, 2018) using sulcal depth as the sole feature. All data were registered

| Optimiser | Learning Rate | Warm-up | Scheduler | PMA | GA |
|---|---|---|---|---|---|
| SGD | $5e^{-3}$ | ✗ | ✗ | $+\infty$ | $+\infty$ |
| SGD | $1e^{-3}$ | ✗ | ✗ | 0.912 | 1.549 |
| SGD | $5e^{-4}$ | ✗ | ✗ | **0.653** | **1.504** |
| SGD | $1e^{-4}$ | ✗ | ✗ | 0.680 | 1.586 |
| SGD | $5e^{-5}$ | ✗ | ✗ | 0.721 | 1.588 |
| Adam | $5e^{-3}$ | ✗ | ✗ | **0.684** | 1.663 |
| Adam | $1e^{-3}$ | ✗ | ✗ | 1.005 | 1.684 |
| Adam | $5e^{-4}$ | ✗ | ✗ | 0.918 | **1.372** |
| Adam | $1e^{-4}$ | ✗ | ✗ | 0.766 | 2.244 |
| Adam | $5e^{-5}$ | ✗ | ✗ | 1.381 | 26.76 |
| SGD | $5e^{-4}$ | ✓50 | ✗ | **0.641** | 1.481 |
| SGD | $5e^{-4}$ | ✓50 | ✗ | 0.714 | - |
| SGD | $5e^{-4}$ | ✓100 | ✗ | 0.643 | - |
| SGD | $5e^{-4}$ | ✓50 | Cosine | **0.645** | - |
| Adam | $5e^{-3}$ | ✓50 | ✗ | 0.844 | 1.518 |
| Adam | $5e^{-3}$ | ✓50 | Cosine | 0.973 | - |
| Adam | $5e^{-4}$ | ✓50 | ✗ | - | **1.437** |

Table 4: Optimising strategy and hyperparameter search for PMA and GA (template aligned data). Mean Absolute Error (MAE) in weeks on the validation set. Models were trained for 1000 iterations.

to a modified version of the 40-week sulcal depth template from the dHCP spatiotemporal cortical surface atlas (Bozek et al., 2018), which was made to be left-right symmetric (Williams et al., 2021). In this paper, template space refers to sphericalised cortical surface data registered to the 40-week template, and native space refers to sphericalised cortical surface data prior to registration. Finally, the cortical data is resampled, using barycentric interpolation, from its template resolution (32492 vertices) to a sixth order icosphere (mesh of 40962 equally spaced vertices).

In all experiments, datasets were group-normalised across feature channels to a mean of 0 and standard deviation of 1, and split into train/validation/test ratio: 80%/10%/10%.

### A.4. Training of gDL methods

The proposed vision transformer framework is benchmarked against five geometric deep learning methods for estimating surface convolutions, including two graph convolutional networks: ChebNet (Defferrard et al., 2017) and GConvNet (Kipf and Welling, 2017); S2CNN (Cohen et al., 2018) - which implements spectral convolutions in $S0(3)$; Spherical-Unet (Zhao et al., 2019), which learns localised filters fit to the hexagonal tessellation of a regular icosphere; and MoNet (Monti et al., 2016), which fits filters as a mixture of isotropic and anisotropic Gaussians. All networks were implemented with 4 convolutional layers, each followed by a ReLU activation and a downsampling operation. Channel size doubled after each convolution from an initial value that was set to 32, for all models except S2CNN which began at 16 due to memory constraints. A fully connected layer was used to make a final age prediction, where for birth age prediction, an additional 1D convolution was used to incorporate scan age as a confound. Data augmentation was implemented with

rotations of the icospheres and non-linear warpings of the icosahedral meshes. Balancing of the datasets between terms and preterms was also implemented. For more details see Fawaz et al. 2021.

## Appendix B. Additional Results

### B.1. Self-Supervision

Various settings for the self-supervision task of *masked patch prediction (MPP)* have been tested and then finetuned for the task of PMA. In Table 5, we report results with and without finetuning the entire model but only the MLP head. Pretraining is evaluated with template data only or both template and native data; and with a probability of masking patches in the sequence of either 25% or 50%. In either configuration, pretrained models are then finetuned with only the MLP head trainable (✗) or all layers (✓). The baseline corresponds to a pretraining task on template data with 50% chance of masking patches in the sequence with only the MLP head finetuned for the PMA task (first row in Table 5).

| Models | Pretraining | Template | Native | Prob.Mask. | Finetuning | Params. | PMA | Δ |
|--------|-------------|----------|--------|------------|------------|---------|-----|---|
| SiT-tiny | dHCP | ✓ | ✗ | 0.5 | ✗ | 577 | 0.691 | base. |
| SiT-tiny | dHCP | ✓ | ✗ | 0.5 | ✓ | 5M | **0.597** | -13.6% |
| SiT-tiny | dHCP | ✓ | ✗ | 0.25 | ✗ | 577 | 0.851 | +23.2 |
| SiT-tiny | dHCP | ✓ | ✗ | 0.25 | ✓ | 5M | 0.817 | +18.2% |
| SiT-tiny | dHCP | ✓ | ✓ | 0.5 | ✗ | 577 | 0.710 | +2.7% |
| SiT-tiny | dHCP | ✓ | ✓ | 0.5 | ✓ | 5M | 0.680 | -1.6% |
| SiT-tiny | ImageNet | ✓ | ✗ | 0.5 | ✗ | 577 | 0.795 | +15.0% |
| SiT-tiny | ImageNet | ✓ | ✗ | 0.5 | ✓ | 5M | 0.675 | -2.3% |

Table 5: Masked patch prediction task. Models are trained for 1000 iterations, finetuning on the task of PMA on template data - results are presented on validation set.

### B.2. Deconfounding strategy

As explained in Section 4, the scans used for term and preterm subjects for the task of birth age prediction (GA) are confounded by appearance of the scans, and therefore the PMA at scan. Deconfounding PMA at scan for this task constitutes a solution to improve prediction results of birth age. A simple framework for deconfounding the data is illustrated in Figure 3, where the PMA at scan per subject is normalised with a BatchNorm layer and then projected via a learnable linear layer to a vector of dimension $D$. The resulting embedding is added to the sequence of patch embeddings. In Table 6, results before and after deconfounding are presented with a *SiT-tiny* model for both template and native data.

## Appendix C. Vision Surface Transformers

### C.1. Multi-Head Self-Attention

Self-attention is the main operation (Eq.2) of the MSA layers. Implemented as:

$$\text{Attention}\left(Q, K, V\right) = \text{Softmax}\left(QK^{\top}/\sqrt{D}\right)V \tag{2}$$

| Methods | Pretraining - MPP | GA | |
|---|---|---|---|
| | | **Native** | **Template** |
| SiT-tiny | ✗ | 1.85 | 1.38 |
| SiT-tiny | ✓ | 1.76 | 1.25 |
| SiT-tiny-deconfounded | ✗ | 1.66 | 1.37 |
| SiT-tiny-deconfounded | ✓ | 1.61 | 1.18 |

Table 6: Best results on birth age (GA) prediction task, with and without deconfounding strategy. Comparison with *SiT-tiny* models trained from scratch and after pretraining.

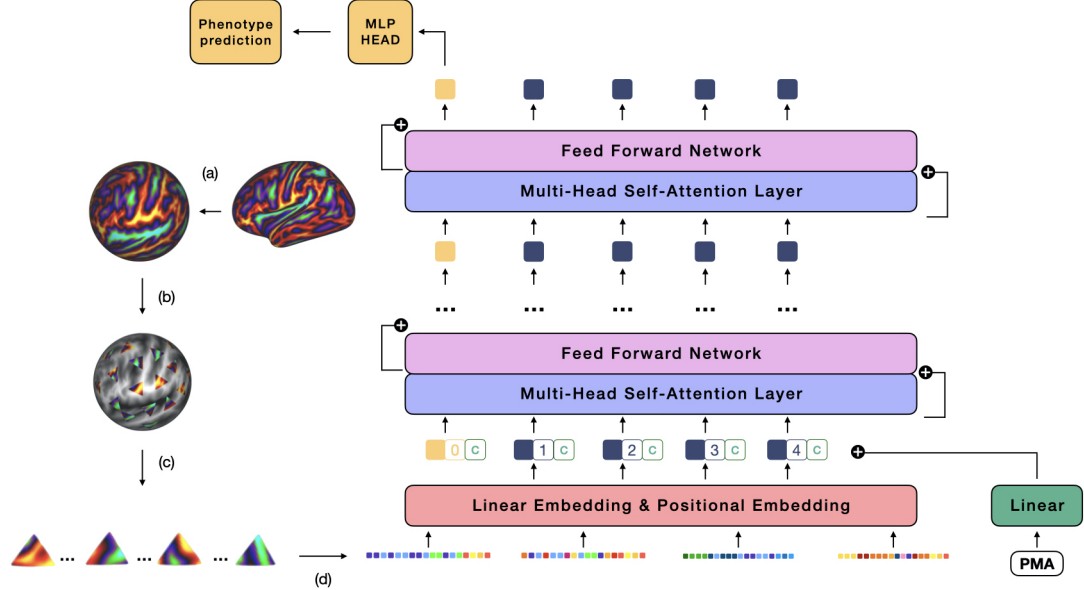

Figure 3: Illustration of the Surface Vision Transformer model implementation a deconfouning strattegy of PMA for the GA prediction task.

it is based on the computation of attention weights between tokens in a sequence. Here, each element of the input sequence is associated with a triplet: the **$Query$**, **$Key$**, and **$Value$** $(Q, K, V \in \mathbb{R}^{N \times D})$ with each computed via linear projection of the input tokens (see Figure 4.a), such that: $Q = XW_Q$, $V = XW_V$, $K = XW_K$. From these, self-attention weights $(\omega_i = [\omega_{i,j}]_{j=1...N})$ are estimated for each patch $i$, from the inner product $\omega_{i,j} = \langle q_i, k_j \rangle$, between the query $(q_i)$ of patch $i$, and keys from all patches $(k_j, \forall j \in [\![1, N]\!])$. From this the self-attention matrix $A = \text{Softmax}\left(QK^\top / \sqrt{D}\right) \in \mathbb{R}^{N \times N}$ is then constructed; normalised (Figure 4.b) and passed through a softmax layer (per row) (Figure 4.c). Finally the output sequence is obtained by weighting the values columns $V$ based on the self-attention weights (Figure 4.d).

## C.2. Positional Embeddings

A positional embedding vector $E \in \mathbb{R}^D$ is added to each patch embedding of the sequence $X^{(0)}$. This vector should encode spatial information about the sequence of patches. The

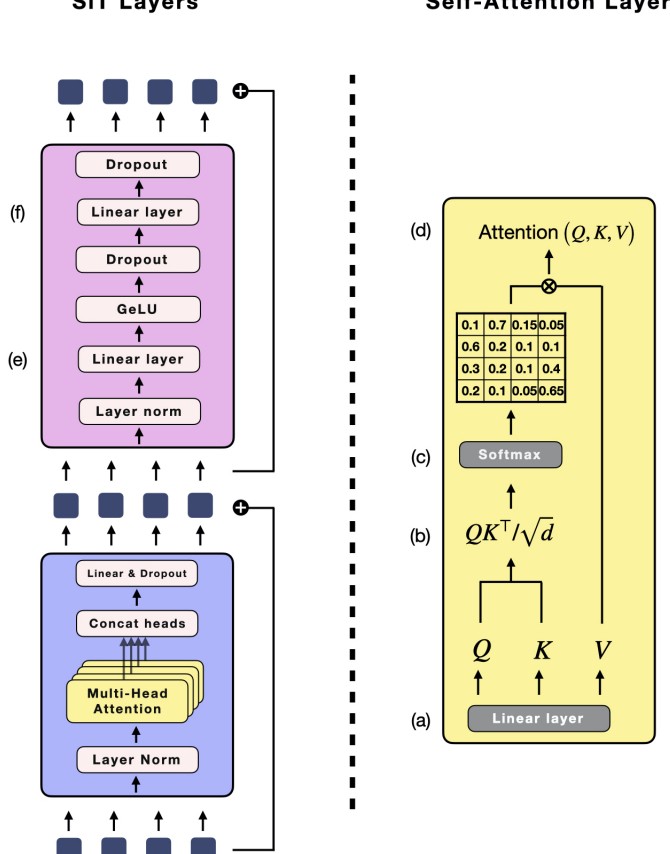

Figure 4: Multi-Head Self-Attention module (MSA) in purple, Self-attention layer in yellow and Feed Forward Network (FFN) Layer in pink. FFN layer expands the sequence dimension to $4D$ (e), then reduces it to $D$ after activation and dropout (f).

$SiT$ implements a positional encoding in the form of a 1D learnable weights, similar to the approach employed in Dosovitskiy et al. 2020.

## C.3. Attention maps

Additional illustrations of attention maps are provided in Figure 5 and Figure 6. Template maps for term and preterm per attention head were created by averaging all attention maps for term and preterm subjects. Attention maps are generated from models trained for the PMA task from scratch Figure 5.a and after self-supervision Figure 5.b, and similarly for the task of GA in Figure 6. Those templates provide valuable insights on both prediction tasks and subjects (see legends in Figure 5 and 6).

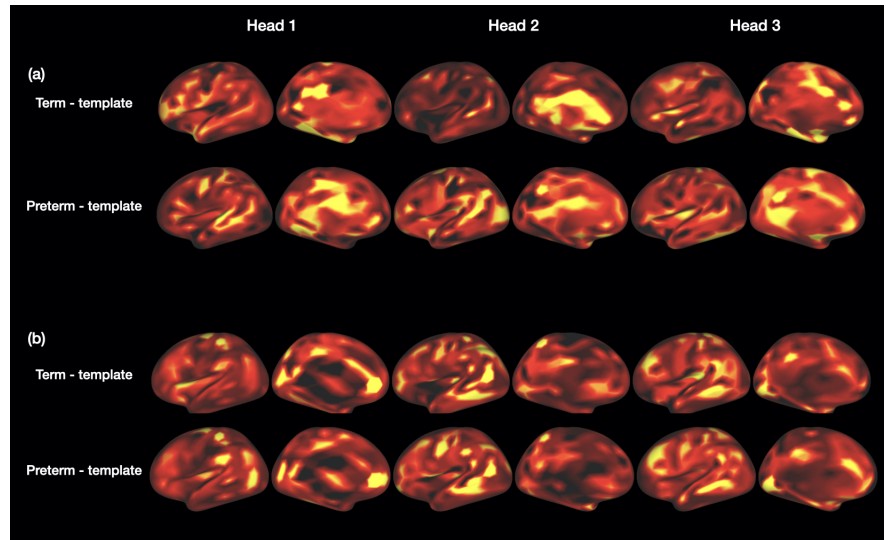

Figure 5: Average attention maps in the PMA prediction task across heads for term and preterm neonates (a) when training from scratch and (b) training after self-supervision. With self-supervision, attention shifts away from the medial wall cut (an artefact of cortical surface processing), towards the lateral cortical surface. Attention is complementary across heads, and is highest in association cortical areas, which is consistent with the primary-to-association trajectory of cortical development in the perinatal period. Moreover, attention maps are similar between preterm and term neonates.

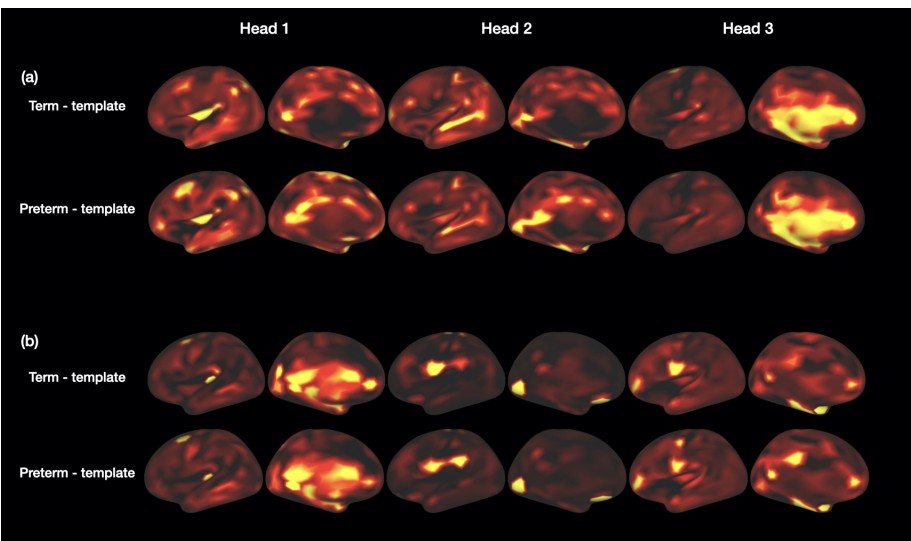

Figure 6: Average attention maps in the GA prediction task across heads for term and preterm neonates (a) when training from scratch and (b) training after self-supervision. Unlike the PMA task, the medial wall cut remains an important feature even in models trained with self-supervision which may be due to the fact that predicting GA is a harder task. Areas of high attention are less common in these average maps, and may reflect that cortical areas important for GA prediction are more variable across subjects.

