# OpenReview forum: "Surface Vision Transformers: Attention-Based Modelling applied to Cortical Analysis"
_MIDL.io/2022/Conference — MIDL 2022_

### Official Review · Reviewer_vMm7 · 2022-01-06

**Confidence:** 4
**Preliminary Rating:** 5
**Recommendation:** Oral

**Summary:**

Good paper describing an implementation of vision transformer on the genus 0 brain surface. This was applied to regress postmenstrual age (PMA) at scan and gestational age (GA) at birth MRI, using T1w and T2w as a proof of concept. Data are from the Developing Human Connectome Project and used close to 600 scans. Regression results were compared to other surface-based deep learning methods, and attention map were qualitatively commented.

**Strengths:**

The adaptation from 2D image (regular grid) to 2D surface (irregular mesh) makes sense as the sphere is parcellated into triangular patches, which are then flattened to feed an (almost) standard ViT.

The paper is well written and clear.

The experimental method is sound with a large dataset.

**Weaknesses:**

No major weakness

One key detail of adapting a flat 2D image to a genus 0 surface is the location encoding, but this is not explained in the paper, unless I missed it. How does the method encode location to guarantee the continuity over the genus 0?

**Deanonymize Review:**

yes

**Detailed Comments:**

1/ The application is convincing, although it is difficult to assess the topological results given the noisy variation across the whole brain (not specific to this method). A better location comparison with standard methods (e,g. VBM or classic regression on the surface) might provide more insight about the advantages of the proposed method to potentially identify cortical areas. Using a classic atrophy study (e.g. Alzheimer’s) would also be interesting for comparison.

2/ When comparing with the other methods, are they all using the same spherical surface inputs (using the same spherical extraction and registration)?

3/ In section 3.2 there seem to be a mistake repeating 2 features: “four input channels (myelin, sulcal depth, curvature, and myelin)”


**Paper Type:**

methodological development

**Questions To Address In The Rebuttal:**

Explain how the location encoding was performed as input to the network.

Clarify and correct the 2 comments above:
- correct section 3.2, I think that one input is cortical thickness.
- clarify the processing pipeline between the various methods, and what steps are common.


**Special Issue:**

yes

---

### Official Review · Reviewer_bYrc · 2022-01-24

**Confidence:** 4
**Preliminary Rating:** 4
**Recommendation:** Best Paper Award, Oral, Poster

**Summary:**

The paper presents a novel model inspired by vision transformers for regression problems on brain surfaces. While some recent papers have used transformers for cortical surface reconstruction and the longitudinal analysis of brain surfaces, this is the first paper to investigate such model for regression tasks.

The model proposed in this paper projects cortical surfaces on a sphere which is divided in a set of triangular patches. Flattened patches are given as input to a transformer based on the data efficient image transformer (DeiT) architecture, along with an extra token for regression. The transformed token at the output of the transformer is used as prediction for a regression task. Self-supervised pre-training is leveraged to improve learning in the model. The proposed method is tested on the tasks of predicting postmenstrual age (PMA) at scan and gestational age (GA) at birth, using cortical surface meshes of neonatal subjects from the Developing Human Connectome Project (dHCP) dataset. Results show that the proposed model offers competitive performance compared to several geometric deep learning approaches for surface analysis.






**Strengths:**

* Novelty: The paper proposes a novel use of transformers for surface-based regression. While inspired by DeiT, the proposed model extends this architecture to genus-zero surfaces by projecting vertices on the surface to an icosphere. As mentioned in the paper,  this generic model could be useful for a broad range of applications involving surface data.

* In-depth validation: The experiments used to validate the proposed method are quite comprehensive. Three architectural variants of different complexity are tested, and two types of pre-training evaluated. The proposed model is compared against five state-of-the-art methods based on geometric deep learning. Additional experiments are presented in the Appendix to further validate the proposed method and its various components.

* Writing: Except for a few minor mistakes, the paper is very well written and easy to follow.




**Weaknesses:**

* Significance: Although I appreciate the novelty of the method, the experiments in the paper do not clearly demonstrate its usefulness compared to existing approaches like MoNet. The prediction accuracy of the proposed model is best for only one case out of four, and computational advantages are unclear.

* Prior work: The review of related works could be improved to better explain the novel contributions of the paper. For instance, authors claim that "there no application of vision transformers to generic functions on surfaces", however one could argue that transformers acting on point clouds, like the Point Transformer by Zhao et al., are in fact capable of surface-based regression. The claim that "geometric deep learning frameworks struggle to learn rotationally equivariant, expressive features without prohibitively high computational cost", should also be better justified. For example, the method by Gopinath et al. uses spectral alignment to account for cross-subject variations in terms of geometry. This method and/or related ones should be discussed in the paper.

* Methodology: Some important steps of the methodology could be more detailed. In particular, the projection of the cortical meshes to the icosphere is not fully described. Is it based on spherical inflation, as in some traditional pipelines? How expensive is that step and how robust is the method to this step?

Gopinath K, Desrosiers C, Lombaert H. Learnable Pooling in Graph Convolution Networks for Brain Surface Analysis. IEEE Transactions on Pattern Analysis and Machine Intelligence. 2020 Oct 2.

**Deanonymize Review:**

no

**Detailed Comments:**

* The paper does a good job at describing the application of transformers and geometric deep learning in machine learning and various computer vision tasks, but provides comparatively few details about their application in medial imaging. For instance, several geometric deep learning approaches have been proposed for brain surface analysis tasks, including age regression.

* The definition of template-aligned data and non-registered (native) data should be clear in the paper, without having to refer to Fawaz et al. 2021. Does native data refer to the actual cortical meshes or their projection on the sphere? Is the registration to the template done in the spherical domain?

* Section 3.2: How is the positional embedding defined for the patches (triangular patches on the icosphere) ?

* Section 3.2: "the right hemisphere is flipped to mirror the left to ensure that sequences from both hemispheres correspond to the
same cortical anatomy". Shouldn't the transformer be invariant to this flipping, or can't you model information about the hemisphere in the positional embedding?

* Table 2: Can you add the standard deviation ?

* Section 4: Why not use self-supervised pre-training on other models? This representation learning technique is not limited to transformers.

* Section 4: "While MoNet does offer transformation invariance, its dependence on Gaussian parametrised filters places some restrictions on filter expressivity; this may explain its poorer performance for GA template". I do not fully understand this argument. MoNet uses a mixture of an arbitrary number of gaussian filters hence it can theoretically model any local geometry. How is this more restrictive than splitting the spherical space in a regular grid of triangle patches?

* Section 4: Runtimes and memory usage should ideally be included in results, since the perfomance of the proposed method is comparable to other approaches like MoNet. Otherwise, advantages are unclear.

* Figure 2: "The attention maps capture differences in the T1w/T2w ratio of the somatosensory (orange box), auditory (white box) and visual
(green arrow) cortices, and changes in sulcal depth in the frontal (purple box) and temporooccipital (magenta box) cortices." This is interesting but is not properly validated, for example using a statistical analysis.

* Conclusion: I understand that space is limited, but having a longer discussion of limitations could improve the paper.

Other comments:

* Section 3.2: "sequence of N flatten patches" -> "sequence of N flattened patches"

* Section 3.2: "(myelin, sulcal depth, curvature and myelin)". myelin is repeated twice.

**Final Rating After The Rebuttal:**

5: Strong Accept

**Justification Of The Final Rating:**

The authors have addressed all my concerns. The method is novel and has a high potential. Experiments have been improved to better highlight its advantages. Overall, the paper is a nice contribution to the medical imaging community.

**Paper Type:**

methodological development

**Questions To Address In The Rebuttal:**

In particular, it is essential to better demonstrate the advantages of the method and explain its contributions with respect to recent work using geometric deep learning or transformers for medical imaging tasks. Moreover, important steps of the methodology such as the registration to the sphere and template could be improved. See weaknesses and comments for a more complete list of points to address.

**Special Issue:**

yes

---

### Meta-Review · Area_Chair_QRCh · 2022-02-19

**Recommendation:** Accept (Oral)
**Confidence:** 5

**Metareview:**

This paper should be accepted.

The reviewers all appreciated the work, and gave constructive comments, which the authors acted on to improve the manuscript and give thorough answers. It's what we love to see in science.

Congratulations to the authors. I'd encourage the authors to try to address some final points, such as the ability to show some measure of variance (even if not across many re-run models but maybe for just some subset of experiment, or over population, etc).

---

### Decision · Program_Chairs · 2022-02-28

Accept